# Inter-monitor reliability and validity of the fibion accelerometers in a laboratory-based study of functional activities

Ashokan Arumugam[1,2,3,4]*, Zaineb Ghannami[1], Mahnaz Mahallati[1], Hala Al-Shams[1], Shaza Ahmed[1], Hager Khaled[1], Maha Hussein[1], Hanan Youssef Alkalih[1], Alexander H.K. Montoye[5‡], Kimberly Clevenger[6‡], Arto J. Pesola[7‡]

1 Department of Physiotherapy, College of Health Sciences, University of Sharjah, Sharjah, United Arab Emirates, 2 Neuromusculoskeletal Rehabilitation Research Group, RIMHS – Research Institute of Medical and Health Sciences, University of Sharjah, Sharjah, United Arab Emirates, 3 Sustainable Engineering Asset Management Research Group, RISE - Research Institute of, Sciences and Engineering, University of Sharjah, Sharjah, United Arab Emirates 4 Adjunct Faculty, Department of Physiotherapy, Manipal College of Health Professions, Manipal Academy of Higher Education, Manipal, Karnataka, India, 5 Department of Integrative Physiology and Health Science, Alma College, Alma, Michigan, United States of America, 6 Department of Kinesiology and Health Science, College of Education and Human Services, Utah State University, Logan, Utah United States of America, 7 Active Life Lab, South-Eastern Finland University of Applied Sciences, Finland; Fibion Inc., Jyväskylä, Finland

☯ These authors contributed equally to this work, and they share second authorship.
‡ AHKM, KC, AJP also contributed equally to this work, and they share last authorship.
* ashokanpt@gmail.com, aarumugam@sharjah.ac.ae

**Editor:** Daniel Boullosa, Universidad de León Facultad de la Ciencias de la Actividad Física y el Deporte: Universidad de Leon Facultad de la Ciencias de la Actividad Fisica y el Deporte, SPAIN

## Abstract

### Background

Accelerometer-based physical activity monitors are designed for placement on several body locations, with popular sites being the wrist, waist/hip, and thigh. Thigh placement is particularly valued for its accuracy in measuring postures and related energy expenditures. However, exploring the outcomes of different thigh wear locations remains an area for further research. This study investigated the inter-monitor reliability of Fibion accelerometers for energy expenditure and activity type classification, and their validity by comparing the Fibion's activity classification against direct observation of a structured activity protocol as the reference standard.

### Methods

Thirty healthy, young adults (aged 21.83±2.80 years; 15 women) wore six Fibion accelerometers on three locations (proximal thigh, 10 cm above the patella, and the front trouser pocket) on both thighs while performing 11 functional activities for 70 minutes in a laboratory setting. Inter-monitor reliability for measuring energy expenditure, activity type and intensity, and accuracy for activity type and intensity classification of the pre-defined functional activities were assessed. Validity was assessed for classifying activity type and intensity.

**Data availability statement:** All relevant data are within the manuscript and its Supporting Information files.

**Funding:** The author(s) received no specific funding for this work.

**Competing interests:** A.J.P. is a co-founder of Fibion Inc. The authors declare that they have no other competing interests.

## Results

Reliability estimates (intraclass correlation coefficients (ICC 3, k)) implied good to excellent inter-monitor reliability for measuring energy expenditure during nearly all activities. Furthermore, energy expenditure measurements were equivalent between accelerometers in both trouser pockets, and proximal and distal thighs within a bound of ±.1.60 kcal/min. However, the right pocket was not equivalent to either the right proximal thigh or the right distal thigh. The mean activity classification accuracy ranged from 87–92% for activity type and 91–94% for activity intensity for the chosen activities, irrespective of the accelerometer location or side of the body (right vs. left).

## Conclusion

The Fibion accelerometers reliably measure energy expenditure and accurately classify activity type/intensity for nearly all functional activities, regardless of the thigh wear location or side of the body. However, interchanging pocket and thigh placements is not recommended. Similar studies in free-living settings are further warranted.

## 1. Introduction

In 2020, the World Health Organization (WHO) updated its physical activity (PA) guidelines, recommending that all adults (aged 18–64 years) do at least 150–300 minutes of moderate- intensity PA per week, 75–150 minutes of vigorous-intensity PA per week, or an equivalent combination for substantial health benefits. For the first time, the guidelines advocate for substituting sedentary time with physical activity of any intensity [1]. According to a systematic review of studies conducted in the Middle East and North Africa (MENA) region in 2021, around 49% of adults are not physically active enough to meet the recommended guidelines for PA. This might be due to the current living environment in several MENA countries that are characterized by technological improvements, lack of suitable sports facilities and hot weather conditions allowing for an increasingly sedentary lifestyle [2].

In the past decade, there has been an increase in the use of accelerometer-based wearable monitors to track sedentary behavior, PA, and sleep. Accelerometers are unobtrusive and are less likely to result in biased estimates often seen with recall methods [3]. Accordingly, acceleration measurements have been used to define the PA profile and its relationship to health, often finding stronger associations between accelerometer-measured PA and health when compared with associations derived using PA recall methods [4].

Accelerometers are designed for placement on several body locations, with popular sites being the wrist, waist/hip, and thigh, and each of these locations come with unique advantages and disadvantages which may preference their use in certain settings. For example, wrist-worn accelerometers are known for being worn with high compliance and allow for measurement of sleep, in addition to PA and sedentary behavior [5]. However, the sedentary and upright time estimates may be less

accurate with wrist-worn accelerometers, particularly during activities like active sitting, standing, or cycling where the hand movement does not correspond with the overall activity level [6–8].

Similar challenges pertain to waist/hip-worn accelerometers, which have relatively low accuracy for identifying sedentary behavior [9–13] but also have lower compliance than wrist or thigh placements [14]. In contrast to the wrist and waist/hip-worn accelerometers, there are several advantages of placing an accelerometer on the thigh. The thigh angle is distinctly different during sitting and upright activities such as standing, and the thigh moves differently during activities such as walking and cycling. Therefore, the thigh- worn accelerometers can precisely differentiate between these activities. Indeed, accelerometer-based estimates of sedentary behavior have reliability ranges from 86–100% when on the thigh, which is superior to that of the wrist- or waist/hip-worn accelerometers [11–13]. Additionally, Montoye et al. discovered that the thigh-worn accelerometers had a high level of accuracy in predicting the duration of each PA intensity category used in their study [15]. These findings further emphasize that the thigh-worn accelerometers may identify sedentary behavior and PA more accurately than the traditional wrist and waist/hip-worn accelerometers [16]. This is because of their accuracy in measuring the postural component of sedentary behavior (i.e., lying, sitting, and reclining postures) alongside PA [13–19]. Finally, through assessment of thigh position, some thigh- worn accelerometers attempt to differentiate between sitting and lying, in order to derive sleep estimates, although currently these are not equivalent to other methods of estimating sleep time [20–22].

The Fibion accelerometer (Fibion Inc, Jyvaskyla, Finland), a pocket- or thigh-worn accelerometer, is designed to assess non-wear time, differentiate between types of activity (sitting, standing, walking, cycling and running/high-intensity), quantify activity intensity (light, moderate, and vigorous), and estimate total and activity specific energy expenditure [23]. Sitting and standing are distinguished based on the Fibion's orientation, whereas the ambulatory activities are differentiated based on the device orientation and the movement pattern typical for these activities [24]. In healthy adults, a previous study found that the Fibion worn on the proximal anterior thigh displayed good to excellent validity in measuring sedentary and upright time compared to the activPAL4 affixed on the anterior mid-thigh [23]. Further, Yang et al. (2018) demonstrated that the Fibion worn on the proximal anterior thigh accurately recognized sitting, different intensities of PA, and energy expenditure. The authors included direct observation (and video recording) as the criterion measure for validating different types and intensities of physical activity. However, the Fibion worn in the front trouser pocket did not give similar accurate outcomes for energy expenditures and PA intensities compared to the Fibion worn on the proximal anterior thigh [25]. Therefore, the choice of thigh or pocket wear of the Fibion monitor may affect PA and sedentary behavior outcomes [25]. Moreover, pocket placement of (Fibion) accelerometers has not been studied extensively. Recently, the Prospective Physical Activity, Sitting, and Sleep consortium (ProPASS) guidelines have recommended the distal anterior thigh as a suitable location for thigh-worn devices, but did not specify whether the right or left side should be used [26]. Studies comparing side-to-side differences in sedentary and upright activity classification and energy expenditure estimation by the Fibion are lacking. Thus, comparing the accuracy and reliability of Fibion across different wear locations on the (ipsilateral and contralateral) thigh will help researchers determine the optimal location for wearability, and comparability with studies that may use various wear locations [16].

The aim of the present study was two-fold: 1. to investigate the inter-monitor reliability of Fibion accelerometers worn in three different locations on both sides: on the proximal thigh, distal thigh, and in a front pants/trouser pocket for assessing energy expenditure and classifying activity type and intensity of predefined functional activities; and 2. to examine Fibion validity (classification accuracy based on activity types and intensities) of the predefined functional activities included in the study.

## 2. Methods

### 2.1. Study design

A laboratory-based study was conducted with healthy young adults, involving a single session lasting approximately 70 minutes. Ethical approval (REC-23-02-23-05-S) was obtained from the Research Ethics Committee, University of Sharjah, United Arab Emirates. The study adhered to the ethical principles included in the Declaration of Helsinki. A written informed consent was obtained from all participants before data collection.

## 2.2. Study setting

The study took place at the College of Health Sciences, University of Sharjah, United Arab Emirates.

## 2.3. Study participants

Thirty healthy adults aged between 18 and 35 years with a body mass index (BMI) >18.5 kg/m$^2$ volunteered to participate in the study between 02 March 2023 and 31 May 2023. Participants were able to ambulate without limitations and did not have any conditions including asthma, musculoskeletal/rheumatic disorders, diabetes mellitus, or cardiovascular or chronic diseases. Participants were invited to participate in the study through emails, word of mouth, and adverts posted on the university notice board and social media.

## 2.4. Sample size estimation

For a minimum acceptable ICC of 0.50, an expected ICC of 0.80, an α error of 0.05, and a power of 0.80, the required sample size was 28. Therefore, 30 participants were considered sufficient for this study [27].

## 2.5. Instrumentation

Fibion accelerometers (Fibion Inc., Jyväskylä, Finland), manufacturer-supplied thigh straps, and Hypafix® tapes were used in this study. The Fibion is a small, lightweight device (20 g, length × width × thickness = 30 × 32 × 10 mm) which connects to the computer via cable and can be started/stopped using the Fibion sync tool application.

## 2.6. Procedure and protocol

To ensure that the participants met the inclusion criteria, they were asked to complete a screening form and the extended version of the Nordic Musculoskeletal Questionnaire (NMQ-E) which gathers information on musculoskeletal pain over the past 12 months and the past 7 days [28]. Study objectives, procedures, and instructions were provided to eligible participants through verbal explanation and an information sheet, and a signed informed consent was sought prior to their participation. Lower limb dominance was recorded as the leg self-preferred by each participant to kick a ball [29]. Participants were asked to wear trousers (i.e., ankle-length pants) with front pockets for the study.

Participant height was measured using a stadiometer. Their BMI, weight, and body composition were determined using a bioelectrical impedance analyzer (multifrequency body composition analyzer MC-780PMA, Tanita Health Equipment Ltd., Hong Kong) [30,31].

Each participant wore a total of 6 Fibion accelerometers on 3 different locations on both thighs (Fig 1): 1) in the front trouser pockets, 2) on the proximal anterior thigh as recommended by the Fibion manufacturer, and 3) on the anterior thigh at 10 cm above the patella (hereafter referred to as distal thigh location) according to the ProPASS guidelines (26). All 6 accelerometers were worn in the same orientation (company logo upright and facing anteriorly). Thigh straps were used for the thigh accelerometers placed 10 cm above the patella for 5 participants and Hypafix® tape for the proximal thigh accelerometers. As two participants complained of slipping of the strap while performing the activities, the Hypafix® tape was used for all 4 thigh-placed accelerometers for the remaining 25 participants.

After placing the accelerometers, participants performed selected functional activities [32] described in Table 1, for ≈70 minutes, a minimum of 2 hours after their last meal. These activities were selected to include different types and intensities to indicate how the accelerometers might perform during various activities of daily living [15]. We used direct observation as the criterion measure to validate the type and intensity (metabolic equivalent (MET) based cut-off values used by the Fibion software) of pre-defined functional activities performed by our participants [25,33,34]. The exact start and stop times of each activity were recorded so that the accuracy and reliability of the Fibion accelerometers could be determined.

Temperature was measured using a thermometer (normal temperature ranges from 36.1 to 37.2°C [35]) before beginning the lab-based functional activities. Participants' heart rate, blood pressure, respiratory rate, and RPE were measured

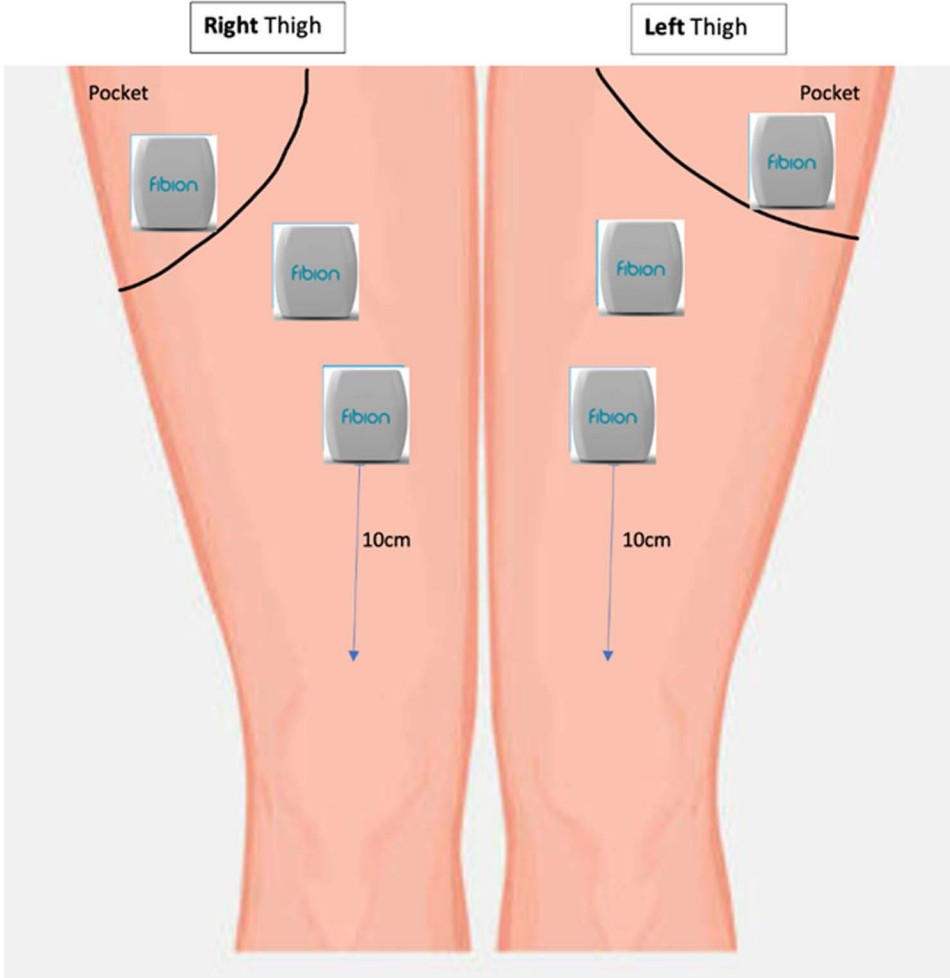

**Fig 1. Fibion wear locations on both thighs.**

before and after each activity. Blood pressure and heart rate were measured by an automated BP monitor (normal blood pressure is < 120 mmHg and <80 mmHg for systolic and diastolic measures, respectively [36], and normal heart rate is 60–100 beats/min [37]), and respiratory rate was measured manually (normal values range from 12 to 20 breaths/min [38]) before and after performing each functional activity included in the protocol. Physical exertion during the activities was monitored with the Borg rating of perceived exertion (RPE), 6–20 scale [39].

While performing strenuous physical activities (e.g., walking on a treadmill at a fast pace), participants were instructed to cease the activity if they reached very hard level on the Borg RPE (17 on the 6–20 scale [39]) or if they felt too uncomfortable to continue the tasks. Participants were permitted to move on to the next activity once their vital signs returned close to baseline values after each activity.

## 2.7. Fibion data processing

Each participant's age, sex, weight, and height were uploaded with the Fibion data to the manufacturer's website (https://beta.fibion.com/upload/research/). The proprietary Fibion cloud data processing tool was used to classify the percent of each 1-minute epoch spent in each activity type and calculate energy expenditure (kcal) overall and by activity type for

**Table 1. Protocol describing the functional activities monitored with the Fibion devices.**

| Activity type based on visual observation | Activity (intensity) | Time (minutes) | Expected Fibion activity type | Expected Fibion intensity |
|---|---|---|---|---|
| Supine lying | Rest period (S) | 5 | Sitting | Sitting |
| Standing | Bipedal, stationary stance without any support (L) | 5 | Standing | Light |
| 6-minute walk test | Brisk, self-paced walk along the corridor for 6 minutes (M/V) | 6 | Walking | Moderate-to-vigorous |
| Sitting | Sit on a chair and type/browse on a computer (S) | 10 | Sitting | Sitting |
| Treadmill* locomotion | 3 km/hour for 1 minute (L)<br>4 km/hour for 1 minute (M)<br>5 km/hour for 1 minute (M)<br>6 km/hour for 2 minutes (M)<br>7 km/hour for 2 minutes (M)<br>8 km/hour for 2 minutes (V)<br>9 km/hour for 2 minutes (V)<br>10 km/hour for 2 minutes (V) | 13 | Walking/running | Moderate-to-vigorous |
| Sitting | Sit on a chair and watch a movie on a laptop (S) | 10 | Sitting | Sitting |
| Stair climbing | Walk up and down a 28-step staircase at a self-selected pace without hand support (M/V) | 3 | Walking | Moderate-to-vigorous |
| Sitting | Sit on a chair and type/browse on a mobile phone (S) | 5 | Sitting | Sitting |
| Cycling | Self-paced cycling at 60–100 revolutions per minute (M/V) | 5 | Cycling | Moderate-to-vigorous |
| Side-lying (left or right) | Rest period (S) | 5 | Sitting | Sitting |
| Shelf organizing | Sorting different weight dumbbells in boxes and putting them in the first shelf of the cupboard, at the level of the trunk (L) | 5 | Standing | Light |

Intensities abbreviated as S: sedentary, L: light, M: moderate, V: vigorous.

*This task includes a combination of walking and running.

each epoch [23–25]. Fibion activity types include non-wear, sitting, standing, walking (a combination of slow and brisk walking), cycling, and high intensity/running. The total energy expenditure is calculated as a sum of activity type energy expenditures and categorized as light (<3 METs excluding sitting) or moderate/vigorous intensity (≥3 METs) activities. Data were then exported as a.CSV file, segmented and summarized to get the total energy expenditure for each of the eleven activities (sitting was repeated thrice). The predominant activity type for each epoch was used for subsequent analysis. Additionally, the amount of time, type, intensity, and energy expenditure were analyzed for each activity performed. Data were extracted based on activity start/stop time using customized R codes. The first- and last- minute data from each activity type were excluded from analysis to prevent overlap between tasks as Fibion data are exported in 1-min epochs.

## 2.8. Statistical analysis

### Aim 1: Inter-monitor reliability analysis of the Fibion accelerometers placed on different locations for energy expenditure, activity intensity, and activity type

Data distribution was confirmed to be normal using the Shapiro-Wilk test. Inter-monitor reliability across the six accelerometers for energy expenditure of each activity and all activities combined was assessed by using the intraclass correlation coefficients (ICC 3, k [two-way mixed, average measure, absolute agreement]), and 95% confidence intervals (CI). Reliability scores were interpreted using the following criteria: poor (< 0.5), moderate (between 0.5 and <0.75), good (between 0.75 and <0.9) or excellent (0.9–1.0), based on 95% CIs of ICC estimates.

Planned comparisons of three right versus left Fibion accelerometer locations (e.g., right versus left pocket) and three right side accelerometer location comparisons (e.g., proximal versus distal thigh) for a combined energy expenditure value of all activities were made with the two one-sided tests (TOST) approach. For these analyses, equivalence bounds were set at 10% of the mean for each variable. For the comparisons to be considered equivalent, p values must be < 0.05.

The reliability of activity type and activity intensity classifications were compared across all six wear locations using Fleiss' kappa (activity type) or Kendall's Coefficient of Concordance (activity intensity because it is ordinal), accelerometers worn on the left versus right side at each location using regular (activity type) or weighted (activity intensity because it is ordinal) kappa, and accelerometers worn on the right side (i.e., pocket versus proximal thigh versus distal thigh), using regular (activity type) or weighted (activity intensity because it is ordinal) kappa. Percent agreement was calculated for all pairwise comparisons.

### Aim 2: Analysis of Fibion validity for classification of activity type and intensity

The Fibion accuracy (percentage agreement) for activity type classification against direct observation and activity intensity based on the categorization summarized in Table 1 was completed separately for each of the six accelerometer locations [40]. All statistical analysis were done using the IBM SPSS software version 21.0 (IBM Corp., Armonk, NY, USA) except equivalence testing, kappa, and Kendall's Coefficient of Concordance, which were done in R Studio (version 2023.06.1) using the TOSTER (version 0.6.0) and irr (version 0.84.1) packages.

## 3. Results

A total of 30 participants (15 females, 15 males), with a mean age of 21.8 ± 2.8 years and BMI of 24.6 ± 5.0 kg/m² completed the study [Table 2]. Among them, three participants self- reported left-sided lower limb dominance while the rest were right-side dominant. Seven participants had missing values for at least one thigh placement due to technical issues (n = 5) or slippage of the Fibion accelerometer when attached with the thigh strap (n = 2). Thus, 30 participants were included with 23 comparisons for each activity for the reliability analysis.

Each participant completed the allocated time assigned for 10 of the 11 activities except for the treadmill locomotion activity, namely those who reached an RPE of ≥17 before the end of the task. The average duration of treadmill locomotion was 11 minutes. Furthermore, the stair climbing and 6MWT activities were self-paced; therefore, some participants performed these activities at a faster rate than others.

### Aim 1: Inter-monitor reliability analysis of the Fibion accelerometers placed on different locations.

The Fibion accelerometers provided reliable energy expenditure values (kcal) for each functional activity carried out in the study irrespective of their wear location (a front trouser pocket, the proximal thigh, or the distal thigh) on both right and left sides (Table 3). Inter- monitor reliability was excellent during standing (ICC = 0.94, 95% CI = 0.90, 0.97), 6MWT (ICC = 0.98, 95% CI = 0.96, 0.99), treadmill locomotion (ICC = 0.98, 95% CI = 0.97, 0.99), stair climbing (ICC = 0.99, 95% CI = 0.99, 1.00), side-lying (ICC = 0.92, 95% CI = 0.85, 0.96) and shelf-organizing (ICC = 0.93, 95% CI = 0.87, 0.97) activities.

**Table 2. Participants' characteristics (n = 30).**

| Variables | Mean (SD) |
|---|---|
| Age (years) | 21.8 (2.8) |
| Height (cm) | 170.6 (9.5) |
| Weight (kg) | 72.3 (18.5) |
| BMI (kg/m²) | 24.6 (5.0) |
| Body Fat (%) | 24.6 (9.2) |
| Muscle Mass (kg) | 50.9 (13.1) |

**Table 3. Intraclass correlation coefficients (ICC 3, K) correlating energy expenditure values (kcal) for different activities estimated by the Fibion accelerometers (n = 23) placed at different locations on both thighs.**

| Activity | Pocket | | Proximal thigh | | Distal thigh | | ICC (95% confidence intervals) | Strength of correlation (reliability) | p-value |
|---|---|---|---|---|---|---|---|---|---|
| | Right | Left | Right | Left | Right | Left | | | |
| | Energy expenditure (kcal) | | | | | | | | |
| Lying | 3.90 ± 0.94 | 4.61 ± 3.21 | 4.54 ± 1.01 | 5.23 ± 3.62 | 3.82 ± 0.56 | 4.79 ± 4.19 | 0.80 (0.65,0.90) | Good | <0.001 |
| Standing | 4.69 ± 0.88 | 5.07 ± 1.13 | 4.70 ± 0.93 | 5.24 ± 0.93 | 4.58 ± 0.87 | 4.93 ± 1.12 | 0.94 (0.90,0.97) | Excellent | <0.001 |
| 6MWT* | 22.28 ± 5.21 | 22.80 ± 6.99 | 23.75 ± 5.49 | 24.13 ± 5.48 | 24.69 ± 4.91 | 25.89 ± 6.88 | 0.98 (0.96,0.99) | Excellent | <0.001 |
| Sitting | 9.78 ± 1.65 | 9.62 ± 1.27 | 11.57 ± 1.98 | 11.52 ± 2.18 | 9.83 ± 1.22 | 10.23 ± 1.67 | 0.54 (0.23,0.77) | Moderate | 0.001 |
| Treadmill locomotion | 81.33 ± 38.77 | 86.92 ± 40.10 | 91.58 ± 40.22 | 88.14 ± 37.71 | 97.32 ± 43.45 | 98.83 ± 42.08 | 0.98 (0.97,0.99) | Excellent | <0.001 |
| Sitting and watching amovie on a laptop | 10.54 ± 1.97 | 11.05 ± 2.11 | 11.96 ± 2.24 | 11.87 ± 2.42 | 10.42 ± 1.76 | 10.58 ± 2.06 | 0.76 (0.57,0.89) | Good | <0.001 |
| Stair climbing (self-paced) | 5.67 ± 1.90 | 5.78 ± 2.19 | 5.94 ± 2.07 | 6.12 ± 2.10 | 6.18 ± 2.11 | 6.29 ± 2.16 | 0.99 (0.99,1.00) | Excellent | <0.001 |
| Sitting and using a mobile phone | 4.16 ± 1.23 | 4.51 ± 1.11 | 4.73 ± 1.06 | 4.71 ± 1.45 | 4.70 ± 1.87 | 4.42 ± 1.43 | 0.88 (0.78,0.94) | Good | <0.001 |
| Cycling | 14.84 ± 3.06 | 14.99 ± 4.07 | 16.54 ± 2.64 | 16.30 ± 2.66 | 18.35 ± 3.04 | 18.64 ± 2.83 | 0.89 (0.76,0.95) | Good | <0.001 |
| Side-lying (left or right) | 3.88 ± 0.86 | 4.08 ± 1.20 | 4.26 ± 1.00 | 4.68 ± 1.36 | 4.12 ± 1.01 | 4.45 ± 0.93 | 0.92 (0.85,0.96) | Excellent | <0.001 |
| Shelf organizing | 5.39 ± 0.95 | 5.40 ± 1.00 | 5.44 ± 0.88 | 5.50 ± 0.81 | 5.16 ± 0.75 | 5.49 ± 0.93 | 0.93 (0.87,0.97) | Excellent | <0.001 |
| All Activities Combined | – | | | | | | 1.00 (1.00,1.00) | Excellent | <0.001 |

*6MWT: 6-minute walk test.

Moreover, good reliability was found for the energy expenditure output values during lying (ICC = 0.80, 95% CI = 0.65, 0.90), sitting while watching a movie on a laptop (ICC = 0.76, 95% CI = 0.57, 0.89), sitting and using a mobile phone (ICC = 0.88, 95% CI = 0.78, 0.94) and cycling (ICC = 0.89, 95% CI = 0.76, 0.95). One activity, namely sitting, showed moderate reliability (ICC = 0.54, 95% CI = 0.23, 0.77; [Table 3]). In addition, the ICCs for all activities (combined) performed in the study was excellent (ICC = 1.00, 95% CI = 1.00, 1.00), indicating excellent reliability of the Fibion accelerometers for measuring energy expenditures across different accelerometer locations.

Equivalence testing results revealed energy expenditure was equivalent between accelerometers on both trousers pockets, proximal and distal thighs within a bound of ± 1.60 kcal/min. However, the right pocket was not equivalent to either right proximal thigh or right distal thigh. The estimate (bias) and confidence intervals for each pair location were - right pocket vs. left pocket: estimate 0.57 (90% CI, 0.20, 0.94); right proximal thigh vs. left proximal thigh: estimate -0.11 (-0.38, 0.17); right distal thigh vs. left distal thigh: 0.24 (-0.14, 0.61); right pocket vs. right proximal thigh: -1.41 (-1.92, -0.89); right pocket vs. right distal thigh: 2.07 (1.36, 2.77); right proximal thigh vs. right distal thigh: -0.60 (-0.19, 1.01).

Reliability for activity type and intensity classifications are reported in Supplementary Tables 1 and 2 in S1 File, respectively. Percent agreement between accelerometers worn at the same location but different sides of the body were >85%, while agreement across the three right- side devices worn at different locations were ≥80%. Fleiss' kappa comparing activity type classification across all wear locations was 0.76 while Kappa values were ≥0.79 for right versus left comparisons and ≥0.71 for right-side comparisons. Kendall's Coefficient of Concordance was 0.67 for overall activity intensity across Fibion accelerometers for all activities combined, while weighted Kappa values were ≥0.71 for right versus left comparisons and ≥0.63 for right-side comparisons.

### *Aim 2: Analysis of Fibion validity for classification of activity type and intensity*

The Fibion accurately classified the activity type of lying as sitting 97–100% of the time. Standing classification accuracy ranged between 71% and 93%. 6MWT was accurately classified as walking 79–93%, while treadmill locomotion was consistently classified as walking/high intensity with 100% accuracy across all wear locations except for left proximal thigh (96.4%). Sitting was accurately classified with 86–100% accuracy, and stair climbing was identified as walking 86–93% of the time. Cycling was correctly classified as cycling 67–100% of the time, while side lying was identified as sitting with 50–97% accuracy. Lastly, shelf organization was correctly classified as standing 66–89% of the time (Table 4).

In evaluating accuracy for intensity classification, lying was correctly classified as sitting 96–100%. Standing as light PA classification ranged from 71–87%. Overground (6MWT) and treadmill locomotion were accurately classified throughout the study (100%). Sitting classification accuracy ranged from 90–100%. Stair climbing classification as moderate to vigorous PA ranged from 93–96%. Cycling classification accuracy as moderate to vigorous PA ranged from 93% to 100%. Side lying classification as sitting ranged from 57–93% and shelf organization was correctly classified 91–94% of the time as light PA (Table 5).

## 4. Discussion

The present study assessed the inter-monitor reliability and validity of Fibion accelerometers worn on the proximal thigh, distal thigh and in a front trouser pocket. The devices were evaluated during various functional activities for their ability to determine energy expenditure and accurately classify activity type and intensity based on their wear locations. Overall, the Fibion accelerometers displayed good to excellent reliability for estimating energy expenditure across almost all activities and locations, with an inter-monitor reliability (ICC 3, K) score of 1.00 for comparisons between right and left sides regardless of wear location. The only moderate reliability was observed during the first sitting task following the 6-minute walk test (6MWT), likely due to participants being restless after engaging in moderate to vigorous physical activity (PA). This restlessness or the movement during the 6MWT might have shifted the accelerometers, particularly those in the pocket, potentially affecting their output. Specific reasoning needs to be elucidated in future research.

**Table 4. Activity classification accuracy (%) by the Fibion for the activities included in the study for different accelerometer locations based on activity type.**

| Activity | Pocket | | Proximal thigh | | Distal thigh | |
|---|---|---|---|---|---|---|
| | Right (n = 30) | Left (n = 29) | Right (n = 30) | Left (n = 28) | Right (n = 28) | Left (n = 28) |
| Lying | 97% | 100% | 100% | 100% | 100% | 100% |
| Standing | 83% | 86% | 83% | 93% | 79% | 71% |
| 6MWT | 87% | 79% | 93% | 89% | 82% | 86% |
| Sitting | 100% | 100% | 100% | 100% | 100% | 100% |
| Treadmill locomotion | 100% | 100% | 100% | 96.4% | 100% | 100% |
| Sitting and watching a movie on a laptop | 97% | 100% | 100% | 96% | 100% | 100% |
| Stair climbing | 93% | 93% | 93% | 86% | 96% | 93% |
| Sitting and using a mobile phone | 93% | 86% | 100% | 100% | 100% | 100% |
| Cycling | 67% | 83% | 97% | 100% | 96% | 86% |
| Side-lying | 73% | 97% | 53% | 50% | 61% | 62% |
| Shelf organizing | 66% | 83% | 79% | 89% | 70% | 79% |
| Mean activity classification accuracy | 87% | 92% | 91% | 91% | 89% | 89% |

**Table 5. Activity classification accuracy (%) by the Fibion for the activities included in the study for different accelerometer locations based on activity intensity.**

| Activity | Pocket | | Proximal thigh | | Distal thigh | |
|---|---|---|---|---|---|---|
| | Right (n = 30) | Left (n = 29) | Right (n = 30) | Left (n = 28) | Right (n = 28) | Left (n = 28) |
| Lying | 97% | 100% | 100% | 100% | 96% | 100% |
| Standing | 87% | 76% | 77% | 86% | 79% | 71% |
| 6MWT | 100% | 100% | 100% | 100% | 100% | 100% |
| Sitting | 100% | 100% | 100% | 100% | 100% | 100% |
| Treadmill locomotion | 100% | 100% | 100% | 100% | 100% | 100% |
| Sitting and watching a movie on a laptop | 97% | 100% | 100% | 96% | 100% | 100% |
| Stair climbing | 93% | 93% | 93% | 96% | 93% | 93% |
| Sitting and using a mobile phone | 97% | 90% | 100% | 100% | 100% | 100% |
| Cycling | 93% | 97% | 100% | 96% | 100% | 100% |
| Side-lying | 90% | 93% | 70% | 61% | 71% | 57% |
| Shelf organizing | 77% | 83% | 87% | 89% | 75% | 82% |
| Mean activity classification accuracy | 94% | 94% | 93% | 91% | 92% | 91% |

Equivalence testing showed that accelerometers worn at the same location on opposite sides (e.g., right vs. left pocket) were within 10% of each other, indicating significant equivalence. However, when comparing different locations on the right side, only the proximal and distal thigh locations were equivalent, not the right pocket. As the energy expenditure estimation was equivalent between the proximal and distal thigh placements, users can consider either location for wearing the Fibion accelerometers without a significant impact on energy expenditure estimation. However, as the trouser pocket placement showed deviations from the thigh positions, the pocket wear and thigh placements are not interchangeable for energy expenditure measurement. Although pocket wear may improve participant's adherence, the lack of equivalence suggests exercising caution while using pocket-worn data interchangeably with those of thigh placements in clinical or research settings when energy expenditure is a key outcome of interest.

For activity type and intensity classification, the Fibion demonstrated high agreement rates. Accelerometers worn at the same location, but opposite sides of the body showed over 85% agreement, and those worn at different locations on the right side (pocket, proximal and distal thigh) showed at least 80% agreement. Kappa values were ≥ 0.71 for right versus left comparisons and ≥ 0.63 for right-side comparisons, indicating good agreement regardless of accelerometer placement on the thigh. This evidence suggests that the Fibion reliably measures most activity types, intensities, and activity-specific energy expenditures for multiple thigh wear locations. Importantly, these findings suggest Fibion can be used according to the ProPASS recommendations.

We found a strong agreement and high accuracy for activity intensity classification across the Fibion accelerometers for all activities combined. Our findings are concordant with a previous study that found that the Fibion worn on the proximal anterior thigh effectively detects activity type and intensity, and estimates energy expenditure during simulated daily activities [25]. Moreover, the Fibion Inc. white paper revealed that irrespective of pocket or thigh wear, the Fibion accurately classified over 90% of the activities carried out in the study [41].

To the best of our knowledge, there is no study to date comparing inter-monitor reliability of energy expenditure simultaneously measured by the Fibion accelerometers placed on multiple thigh locations. However, in close agreement with our study findings, Montoye et al. (2022) found minimal differences between accelerometers worn on the right and left midthighs in a free-living comparison of the Fibion accelerometer [42].

Furthermore, activity classification accuracy was completed for each accelerometer placement for all 30 participants. The Fibion accuracy for activity classification based on activity type showed predominantly good to excellent accuracy for lying, sitting, standing, treadmill locomotion, cycling, and stair climbing activities with an average of 98.17%. Side- lying and shelf organizing activities had somewhat lower accuracy ranging between 50–97%. Side lying activity outcomes showed less than 80% accuracy for most wear locations. This could have occurred because of a lack of Fibion's algorithm to identify and classify side lying. Also, participants held their knees at different angles (full extension to various degrees of flexion), and some might have moved their legs while performing the activity. Some accelerometers identified side lying as standing instead of sitting based on their thigh orientation.

Lower accuracy for the shelf-organizing activity could be due to some participants being more stationary while performing the activity than others which resulted in categorizing some of them as sitting instead of standing, and this also applies for standing being categorized as sitting for some placements. Moreover, it is possible that the outcomes of right and left pocket placements for cycling were affected by the hip flexion accompanied by the type of pocket of each participant (loose fitting or longer in length) during the activity which could have allowed the accelerometers to move throughout the protocol and thus not recording accurate measurements.

Accuracy for activity intensity classification was high during lying, 6MWT, treadmill locomotion, sitting, stair climbing, and cycling activities with classification accuracies consistently above 90%. Standing, side lying, and shelf organizing had somewhat lower accuracies likely due to the same reasons mentioned above resulting in categorizing standing and shelf organizing as sitting instead of light physical activity, and side lying as light physical activity instead of sitting. Additionally, the later in the protocol, there may have been a greater likelihood that the pocket-worn accelerometers could have shifted, which could explain the lower accuracy for classifying the side-lying and shelf-organizing activities, which were the last two activities performed by the participants in this protocol. Overall, the Fibion accuracy for activity classification analysis showed more accurate outcomes in classifying activities based on activity intensity than activity type. Moreover, there were no significant differences based on wear location or leg dominancy in this analysis. In agreement with our findings, Montoye et al. (2022) reported that a Fibion accelerometer worn on the right thigh had good accuracy for recognizing sitting, standing, and ambulating activities [42]. Moreover, as per prior studies, the Fibion has been found to be a valid accelerometer in categorizing sitting, standing, walking, cycling, high-intensity movements, and typical daily activities [23,32].

### 4.1. Strengths, methodological considerations, and limitations of the study

The current study is the first of its kind as it compares three different accelerometer placements (a front trouser pocket, proximal anterior thigh, and distal anterior thigh) on both thighs (ipsilateral and side-to-side comparisons included). Furthermore, our study used the recent recommendation from the ProPASS guidelines for the distal thigh accelerometer placement (10 cm above the patella) to standardize the distal thigh Fibion wear position amongst participants enhancing reliability of results. Additionally, the accuracy of Fibion classification for various activity types was able to be verified within the pre-defined functional activities.

Study limitations should also be considered. First, the preliminary implementation of thigh straps on the distal thigh led to missing data in two participants for this location as the accelerometer was recurringly slipping off and as a result was discontinued. In lieu of this, Hypafix® tape was applied to the remaining participants instead. It remains to be determined if or how the thigh straps might be used with the Fibion devices, but our experience suggests that taping the device will result in a more consistent device wear location. With regards to the side-lying activity it was not specified which side participants should lie on, and the position of the knee was not specified to participants whether to remain flexed or extended during the study; this might have affected accelerometer output due to differing orientations, and further the Fibion does not include side-lying or lying down in its outputs. This precludes use of the Fibion for estimating sleep behavior with the current algorithm. As this study took place in a laboratory setting and with a fairly homogeneous sample of young, healthy adults, our results should be confirmed in other populations as well as during free living and for longer time periods, especially as some of the misclassification of activities from the pocket- worn accelerometer may have been due to shifting in the pocket, which could get worse the longer the accelerometer is worn. However, we did not control for trouser tightness because in real life scenarios individuals naturally wear trousers with varying pocket sizes and levels of tightness. We were unable to assess the validity of energy expenditure outcomes, as we used direct observation as the criterion measure. Future studies could use indirect calorimetry to validate those outcomes.

## 5. Conclusion

The Fibion accelerometer demonstrated good to excellent inter-monitor reliability of accelerometer-measured energy expenditure values for 10 out of the 11 activity types irrespective of the thigh wear location. Also, the Fibion accelerometers demonstrated accurate classification of most activities included in the study based on activity type and intensity of the tasks performed. Thus, our findings suggest that the Fibion accelerometer is a reliable and valid sedentary behavior and PA measurement tool irrespective of the wear location (the front trouser pocket, proximal or distal thigh) to measure energy expenditure and identify sitting and different types of physical activities in healthy adults. Even so, pocket wear and thigh placements are not interchangeable for energy expenditure measurement. Future studies on the inter-monitor reliability and accuracy of the Fibion accelerometers worn at different locations on the thigh under free-living settings are warranted.

## Supporting information

**S1 File. Supplementary Table 1**: Reliability of activity type classification accuracy by the Fibion accelerometer for all activities combined for different accelerometer locations and same-side comparisons. **Supplementary Table 2**: Reliability of activity intensity classification by the Fibion accelerometer for all activities combined included in the study for different accelerometer locations and same-side comparisons.
(PDF)

**S3 Data. Raw datasets used for reliability and/or validity analysis of activity type and intensity classification by the Fibion accelerometer. Includes data from all accelerometer locations and tasks analyzed in the study.**
(XLSX)

## Acknowledgments

The authors would like to thank all participants who took part in this study.

## Author contributions

**Conceptualization:** Ashokan Arumugam.

**Data curation:** Ashokan Arumugam, Zaineb Ghannami, Mahnaz Mahallati, Hala Al-Shams, Shaza Ahmed, Hager Khaled, Maha Hussein, Hanan Youssef Alkalih, Alexander H.K. Montoye, Kimberly Clevenger, Arto J. Pesola.

**Formal analysis:** Ashokan Arumugam, Zaineb Ghannami, Mahnaz Mahallati, Hala Al-Shams, Shaza Ahmed, Hager Khaled, Maha Hussein, Kimberly Clevenger.

**Funding acquisition:** Ashokan Arumugam.

**Investigation:** Ashokan Arumugam, Zaineb Ghannami, Mahnaz Mahallati, Hala Al-Shams, Shaza Ahmed, Hager Khaled, Maha Hussein, Hanan Youssef Alkalih.

**Methodology:** Ashokan Arumugam, Zaineb Ghannami, Mahnaz Mahallati, Hala Al-Shams, Shaza Ahmed, Hager Khaled, Maha Hussein, Hanan Youssef Alkalih, Alexander H.K. Montoye, Kimberly Clevenger, Arto J. Pesola.

**Project administration:** Ashokan Arumugam.

**Resources:** Ashokan Arumugam.

**Software:** Ashokan Arumugam, Kimberly Clevenger.

**Supervision:** Ashokan Arumugam, Hanan Youssef Alkalih.

**Validation:** Ashokan Arumugam, Zaineb Ghannami, Mahnaz Mahallati, Hala Al-Shams, Shaza Ahmed, Hager Khaled, Maha Hussein, Hanan Youssef Alkalih, Alexander H.K. Montoye, Kimberly Clevenger, Arto J. Pesola.

**Visualization:** Ashokan Arumugam, Zaineb Ghannami, Mahnaz Mahallati, Hala Al-Shams, Shaza Ahmed, Hager Khaled, Maha Hussein, Hanan Youssef Alkalih, Alexander H.K. Montoye, Kimberly Clevenger, Arto J. Pesola.

**Writing – original draft:** Ashokan Arumugam, Zaineb Ghannami, Mahnaz Mahallati, Hala Al-Shams, Shaza Ahmed, Hager Khaled, Maha Hussein.

**Writing – review & editing:** Ashokan Arumugam, Hanan Youssef Alkalih, Alexander H.K. Montoye, Kimberly Clevenger, Arto J. Pesola.

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
