## [Decision Letter · Decision Letter 0]

7 Mar 2025

PONE-D-24-50753Inter-monitor Reliability and Validity of the Fibion Accelerometers Worn on the Thigh and in a Front Trouser Pocket During Functional ActivitiesPLOS ONE

Dear Dr. Arumugam,

Thank you for submitting your manuscript to PLOS ONE. After careful consideration, we feel that it has merit but does not fully meet PLOS ONE’s publication criteria as it currently stands. Therefore, we invite you to submit a revised version of the manuscript that addresses the points raised during the review process.

**There is strong disagreement between the reviewers. However, the concerns of reviewer #2 are indeed justified. In order to avoid wasting more time waiting for a third experienced reviewer who is difficult to find, I would like to ask the authors to address ALL the concerns of the reviewers. Please note that addressing all issues does not guarantee acceptance of the paper. If the authors moderate the message and really present and discuss the objective findings, the paper may be considered further.**

We look forward to receiving your revised manuscript.

Kind regards,

Daniel Boullosa

Academic Editor

PLOS ONE

Reviewers' comments:

Reviewer's Responses to Questions

**Comments to the Author**

1. Is the manuscript technically sound, and do the data support the conclusions?

Reviewer #1: Yes

Reviewer #2: No

2. Has the statistical analysis been performed appropriately and rigorously? 

Reviewer #1: Yes

Reviewer #2: No

3. Have the authors made all data underlying the findings in their manuscript fully available?

Reviewer #1: No

Reviewer #2: No

4. Is the manuscript presented in an intelligible fashion and written in standard English?

Reviewer #1: Yes

Reviewer #2: Yes

5. Review Comments to the Author

Reviewer #1: I would like to thank the authors and editor for the opportunity to review this manuscript: "Inter-monitor Reliability and Validity of the Fibion Accelerometers Worn on the Thigh and in a Front Trouser Pocket During Functional Activities." The manuscript is well-written, the study design is straightforward yet elegant, the data analysis is robust, and the authors are careful not to overstate their conclusions. With a few minor edits, I believe this manuscript will be ready for publication and will make a valuable contribution to the wearable technology literature.

Line 123: The period should be placed after the citation. Please review the rest of the manuscript to ensure this error is corrected throughout.

The authors need to establish what constitutes a meaningful difference between the proximal and distal placement of the Fibion on the thigh. While it is clear that the use of the front pocket as a wear site is of interest, since valid and reliable data from this location could improve adherence and usability, it is important to clarify what meaningful differences exist between the two thigh wear locations.

Results: Males and females often wear pants with different levels of tightness. If female participants wore tighter pants, this could reduce movement of the Fibion compared to looser pockets. Did the authors consider analyzing the data by sex? If not, what was the rationale for this decision?

Did any participants report instances of the Fibion worn in the pocket coming into contact with the devices secured on the thighs? If so, could this have affected the data collected by either device?

Reviewer #2: The paper aims to assess the reliability and validity of an accelerometer. The study is lacking a clear rational and the methodology is poorly described. While the reliability analysis is somewhat described in the paper, the validity analysis remains unclear. No information concerning the criterion measure was provided. It is assumed that validity was assessed by comparing different accelerometers. However, the added value of this approach is poor the comparison to a gold standard is lacking. The methodology is poorly described and a justification for the selected laboratory protocol was not provided. The reached conclusions are not supported by the data as the statistical approach is insufficiently described.

In addition, I provide a few detailed comments:

Line s72-77: This paragraph appears to be a very simplified. In my view, also the challenges that are brought about by using accelerometers should be pointed out here.

Lines 78 – 103: I am missing the role of different manufacturers here. In line with this, it remains unclear how the Fibion device differs from other models and, hence, why this study is actually needed. Is the motivation of the study to test specifically the accuracy of the Fibion device or is it rather methodological in terms of testing different measurement locations?

Lines 139 – 142: This is a very crucial information that is currently lacking in the title of the manuscript. In fact, including short-term laboratory measures only is a limitation of the study and should be mentioned in the title.

Line 2.1: I am missing a statement concerning the adherence to GCP or the declaration of Helsinki.

Line 162 – 165: What about the criterion used in this study?

Lines 175 – 184: It is unclear why these variables were measured and at what exact time point. I suggest to insert a section describing the study protocol first before going into the details of measurement procedures.

Line 195: What is meant by “after wearing”. I assume you mean after “placing” or something similar. Yet again, the actual protocol used remains unclear.

Lines 227-234: It is unclear for which exact comparisons the ICCs were calculated. Also, the terminology is not clear as you are at most assessing intra-day reliability. However, this is not specified in the manuscript.

Lines 252-253: What do you mean by direct observation? How can energy expenditure be assessed by direct observation? In addition, the statistical approach for assessing validity is insufficient (at least insufficiently described and can, thus, not be judged).

6. PLOS authors have the option to publish the peer review history of their article (what does this mean? ). If published, this will include your full peer review and any attached files.

**Do you want your identity to be public for this peer review?** For information about this choice, including consent withdrawal, please see our Privacy Policy .

Reviewer #1: No

Reviewer #2: No

---

## [Author Response · Author response to Decision Letter 1]

24 Mar 2025

Dear Editor and Reviewers,

We would like to sincerely thank the editor and the reviewers for their constructive feedback on our manuscript, "Inter-monitor Reliability and Validity of the Fibion Accelerometers Worn on the Thigh and in a Front Trouser Pocket During Functional Activities" (PONE-D-24-50753). We appreciate the time and effort spent on the thorough review, and we believe that the comments have significantly helped improve the quality and clarity of our manuscript. Below, we provide a detailed point-by-point response to each of the comments, outlining the changes made accordingly. Review comments are denoted in colored typing and the corresponding replies are in normal text. Changes made to the text in the manuscript have been highlighted in yellow.

Response to Editor's Comment:

There is strong disagreement between the reviewers. However, the concerns of reviewer #2 are indeed justified. In order to avoid wasting more time waiting for a third experienced reviewer who is difficult to find, I would like to ask the authors to address ALL the concerns of the reviewers. Please note that addressing all issues does not guarantee acceptance of the paper. If the authors moderate the message and really present and discuss the objective findings, the paper may be considered further.

Response: We acknowledge the strong disagreement between the reviewers and appreciate your request to address all concerns raised. We have carefully considered and incorporated all feedback, ensuring that our discussion and conclusions are well-balanced, objective, and appropriately supported by data. We are grateful for the opportunity to revise and resubmit our manuscript based on reviewers’ feedback.

Reviewer #1: I would like to thank the authors and editor for the opportunity to review this manuscript: "Inter-monitor Reliability and Validity of the Fibion Accelerometers Worn on the Thigh and in a Front Trouser Pocket During Functional Activities." The manuscript is well-written, the study design is straightforward yet elegant, the data analysis is robust, and the authors are careful not to overstate their conclusions. With a few minor edits, I believe this manuscript will be ready for publication and will make a valuable contribution to the wearable technology literature.

Response: Thank you very much for your positive feedback and appreciation. Please see our responses below to your comments.

Comment 1: Line 123: The period should be placed after the citation. Please review the rest of the manuscript to ensure this error is corrected throughout.

Response: Thank you for pointing this out. Now we have corrected this error.

Revision:

Line 129: “……….location for thigh-worn devices [27].”

Comment 2: The authors need to establish what constitutes a meaningful difference between the proximal and distal placement of the Fibion on the thigh. While it is clear that the use of the front pocket as a wear site is of interest, since valid and reliable data from this location could improve adherence and usability, it is important to clarify what meaningful differences exist between the two thigh wear locations.

Response: We acknowledge the importance of clarifying the meaningful differences between these placements. Thank you for bringing this to our attention. Please see lines 128 to 134 explaining why we used both thigh locations and lines 309 to 314 explaining the differences in energy expenditure between trouser pocket and thigh placements of monitors. The bias and confidence interval estimates suggest significant differences between the right pocket placement and proximal or distal thigh placements. The ± 1.60 kcal/min provides an objective criterion of equivalence. Since, the right pocket’s energy expenditure measurements did not fall within this bound relative to the thigh positions, this implies a discrepancy. Thus, pocket placement may introduce systematic bias in energy expenditure compared to those of thigh placements.

Revision:

Please see lines 362-370: “As the energy expenditure estimation was equivalent between the proximal and distal thigh placements, users can consider either location for wearing the Fibion accelerometers without a significant impact on energy expenditure estimation. However, as the trouser pocket placement showed deviations from the thigh positions, the pocket wear and thigh placements are not interchangeable for energy expenditure measurement. Although pocket wear may improve participant’s adherence, the lack of equivalence suggests exercising caution while using pocket-worn data interchangeably with those of thigh placements in clinical or research settings when energy expenditure is a key outcome of interest.”

Comment 3: Results: Males and females often wear pants with different levels of tightness. If female participants wore tighter pants, this could reduce movement of the Fibion compared to looser pockets. Did the authors consider analyzing the data by sex? If not, what was the rationale for this decision?

Response: Thank you for this comment. Due to cultural reasons and institutional bylaws, all participants (both males and females) were required to wear only full-length trousers, and it is uncommon for female students to wear overly tight trousers in public. Please see lines 181-182 “Participants were asked to wear trousers (i.e., ankle-length pants) with front pockets for the study.” However, we did not control for trouser tightness because in real life scenarios individuals naturally wear trousers with varying pocket sizes and levels of tightness.

The reliability study follows a within-subject design, meaning each participant serves as their own control. This minimizes inter-participant variability and ensures that differences in measurement reliability are assessed based on device placement rather than sex-based differences. As reliability is assessed within the same individual, differences between males and females do not influence the findings, as each participant’s repeated measurements are compared against their own data. Our inter-monitor reliability study involved testing whether different placements (pocket, proximal thigh, and distal thigh) yield consistent measurements within the same individual across repeated trials. However, future studies could test the impact of different pocket sizes and trouser tightness on the accelerometer recorded data.

Revision:

Lines 450-456: “As this study took place in a laboratory setting and with a fairly homogeneous sample of young, healthy adults, our results should be confirmed in other populations as well as during free living and for longer time periods, especially as some of the misclassification of activities from the pocket- worn accelerometer may have been due to shifting in the pocket, which could get worse the longer the accelerometer is worn. However, we did not control for trouser tightness because in real life scenarios individuals naturally wear trousers with varying pocket sizes and levels of tightness.”

Comment 4: Did any participants report instances of the Fibion worn in the pocket coming into contact with the devices secured on the thighs? If so, could this have affected the data collected by either device?

Response: Thank you for raising this point. None of our participants reported direct contact between the devices. Moreover, in reliability studies involving accelerometers, it is common to assess inter-device reliability by comparing measurements from multiple devices placed at the same location or in proximity (please see some references below). This approach helps determine the consistency between different units under identical conditions. No interference has been expected or reported between the devices (accelerometers) of the same or different brands in these studies.

Alkalih HY, Pesola AJ, Arumugam A. A new accelerometer (Fibion) device provides valid sedentary and upright time measurements compared to the ActivPAL4 in healthy individuals. Heliyon. 2022;8(10):e11103.

Montoye AHK, Coolman O, Keyes A, Ready M, Shelton J, Willett E, et al. Evaluation of Two Thigh-Worn Accelerometer Brands in Laboratory and Free-Living Settings. J Meas Phys Behav. 2022;5(4):233–41.

Crowley P, Skotte J, Stamatakis E, Hamer M, Aadahl M, Stevens ML, Rangul V, Mork PJ, Holtermann A. Comparison of physical behavior estimates from three different thigh-worn accelerometers brands: a proof-of-concept for the Prospective Physical Activity, Sitting, and Sleep consortium (ProPASS). International Journal of Behavioral Nutrition and Physical Activity. 2019;16:1-7.

Reviewer #2:

Comment 1: The paper aims to assess the reliability and validity of an accelerometer. The study is lacking a clear rational and the methodology is poorly described. While the reliability analysis is somewhat described in the paper, the validity analysis remains unclear. No information concerning the criterion measure was provided. It is assumed that validity was assessed by comparing different accelerometers. However, the added value of this approach is poor the comparison to a gold standard is lacking. The methodology is poorly described and a justification for the selected laboratory protocol was not provided. The reached conclusions are not supported by the data as the statistical approach is insufficiently described.

Response: Thank you for your thorough review and feedback on our manuscript.

• We clarified the study aim in the abstract to clearly include the direct observation as the criterion measure

• Our scientific justification and rationale for the study are explained in the introduction section from lines 75 to 134.

• Possible criterion measures for such a study include direct observation, video recording, or indirect calorimetry, or previously validated methods (Wu et al. 2022). Lyden et al. (2014) has recommended direct observation as a criterion measure for validation of physical activity and sedentary behavior. Previous study by Yang et al. (2018) investigating the reliability and validity of Fibion accelerometers included direct observation and video recording, in addition to indirect calorimetry as the criterion measures. In our current study, we used direct observation as the criterion measure.

Please see our response to comment 5 below. “Direct observation” refers to manually observed and recorded task durations and type of activities performed to aid in comparing and validating accelerometer recorded activity types and intensities. Although the energy expenditure cannot be directly calculated by direct observation, validating the type and duration of tasks are important. The classification of intensity types was based on the metabolic equivalents (METs).

Lyden K, Petruski N, Mix S, Staudenmayer J, Freedson P. Direct observation is a valid criterion for estimating physical activity and sedentary behavior. Journal of Physical Activity and Health. 2014 May 1;11(4):860-3.

Wu Y, Petterson JL, Bray NW, Kimmerly DS, O’Brien MW. Validity of the activPAL monitor to measure stepping activity and activity intensity: A systematic review. Gait & Posture. 2022;97:165-73.

Yang Y, Schumann M, Le S, Cheng S. Reliability and validity of a new accelerometer-based device for detecting physical activities and energy expenditure. PeerJ. 2018 Oct 11;6:e5775.

• While reliability analysis was conducted for energy expenditure estimates as well as activity type and intensity classification, our validity analysis (direct observation vs. classification) focused on comparing activity type and intensity classification.

• We have described the methodology in detail (please see lines 143-271) and have added justification for the functional tasks (please see lines 197-200) used in the study.

• Please see our responses to comments 7 and 8 below regarding the test protocol and multiple appropriate statistical analysis methods used. Indeed, the conclusions are supported by our data analysis and relevant findings. However, we have added one more sentence in the conclusion considering the discrepancy between trouser pocket wear and thigh placement data.

• A limitation of study is that we were unable to assess the validity of energy expenditure outcomes, as we used direct observation as the criterion measure. Future studies could use indirect calorimetry to validate those outcomes.

Revision:

Lines 38–41 (Abstract): “This study investigated the inter-monitor reliability of Fibion accelerometers for energy expenditure and activity type classification, and their validity by comparing the Fibion’s activity classification against direct observation of a structured activity protocol as the reference standard.”

Lines 47–48 (Abstract): “Validity was assessed for classifying activity type and intensity.”

Lines 53-54 (Abstract): “However, the right pocket was not equivalent to either right proximal thigh or right distal thigh.”

Lines 59-60 (Abstract): “However, interchanging pocket and thigh placements is not recommended.”

Lines Lines 117-121: “Further, Yang et al. (2018) demonstrated that the Fibion worn on the proximal anterior thigh accurately recognized sitting, different intensities of PA, and energy expenditure. The authors included direct observation (and video recording) as the criterion measure for validating different types and intensities of physical activity. However, the Fibion worn in the front trouser pocket did not give similar accurate outcomes for energy expenditures and PA intensities compared to the Fibion worn on the proximal anterior thigh (25).”

Lines 136-141: “The aim of the present study was two-fold: 1. to investigate the inter-monitor reliability of Fibion accelerometers worn in three different locations on both sides: on the proximal thigh, distal thigh, and in a front pants/trouser pocket for assessing energy expenditure and classifying activity type and intensity of predefined functional activities; and 2. to examine Fibion validity (classification accuracy based on activity types and intensities) of the predefined functional activities included in the study.”

Lines 200-203: “We used direct observation as the criterion measure to validate the type and intensity (metabolic equivalent (MET) based cut-off values used by the Fibion software) of pre-defined functional activities performed by our participants(25,33,34).”

Lines 456-458: “We were unable to assess the validity of energy expenditure outcomes, as we used direct observation as the criterion measure. Future studies could use indirect calorimetry to validate those outcomes.”

Lines 468-470: “Even so, pocket wear and thigh placements are not interchangeable for energy expenditure measurement. Future studies……….”

Comment 2: Lines 78 – 103: I am missing the role of different manufacturers here. In line with this, it remains unclear how the Fibion device differs from other models and, hence, why this study is actually needed. Is the motivation of the study to test specifically the accuracy of the Fibion device or is it rather methodological in terms of testing different measurement locations?

Response: The Fibion Device contains a tri-axial accelerometer, equipped with firmware algorithms that processes accelerometer data, instantly translating it into categorized activity classes and corresponding energy expenditures (Fibion Inc. white paper (2015)). The Fibion has shown good to excellent validity against the ActivPAL device with respect to posture measurement, and similar reliability and validity with Actigraph with respect to energy expenditure estimation. The Fibion and activPAL monitors have been found to have comparable intermonitor reliability. While the ActivPAL has been mostly used to measure postures (sitting, standing, and stepping), the Fibion shows good validity in measuring also higher intensity activities as compared with the Actigraph device (Yang et al. (2018); Alkalih et al. (2022)). Also, the Fibion has been already used to validate the ATLS-2 physical activity questionnaire (Arumugam et al. 2024) and International Physical Activity Questionnaire – Short Form (IPAQ-SF) (Arumugam et al. 2024) in adolescents and/or young adults in the UAE. A summary of studies on the validation of the Fibion devices is available at https://web.fibion.com/publications/

Please see lines 124-134: The choice of thigh or pocket wear o

---

## [Decision Letter · Decision Letter 1]

28 Apr 2025

Inter-monitor Reliability and Validity of the Fibion Accelerometers in a Laboratory-Based Study of Functional Activities

PONE-D-24-50753R1

Dear Dr. Arumugam,

We’re pleased to inform you that your manuscript has been judged scientifically suitable for publication and will be formally accepted for publication once it meets all outstanding technical requirements.

Kind regards,

Daniel Boullosa

Academic Editor

PLOS ONE

Additional Editor Comments (optional):

Reviewers' comments:

Reviewer's Responses to Questions

**Comments to the Author**

1. If the authors have adequately addressed your comments raised in a previous round of review and you feel that this manuscript is now acceptable for publication, you may indicate that here to bypass the “Comments to the Author” section, enter your conflict of interest statement in the “Confidential to Editor” section, and submit your "Accept" recommendation.

Reviewer #1: All comments have been addressed

Reviewer #2: All comments have been addressed

2. Is the manuscript technically sound, and do the data support the conclusions?

Reviewer #1: Yes

Reviewer #2: Partly

3. Has the statistical analysis been performed appropriately and rigorously? 

Reviewer #1: Yes

Reviewer #2: Yes

4. Have the authors made all data underlying the findings in their manuscript fully available?

Reviewer #1: Yes

Reviewer #2: No

5. Is the manuscript presented in an intelligible fashion and written in standard English?

Reviewer #1: Yes

Reviewer #2: Yes

6. Review Comments to the Author

Reviewer #1: (No Response)

Reviewer #2: (No Response)

7. PLOS authors have the option to publish the peer review history of their article (what does this mean? ). If published, this will include your full peer review and any attached files.

**Do you want your identity to be public for this peer review?** For information about this choice, including consent withdrawal, please see our Privacy Policy .

Reviewer #1: No

Reviewer #2: No

---

## [Editor Report · Acceptance letter]

PONE-D-24-50753R1

PLOS ONE

Dear Dr. Arumugam,

I'm pleased to inform you that your manuscript has been deemed suitable for publication in PLOS ONE. Congratulations! Your manuscript is now being handed over to our production team.

Kind regards,

on behalf of

Dr. Daniel Boullosa

Academic Editor

PLOS ONE